# Spatial and Temporal Changes in PD-L1 Expression in Cancer: The Role of Genetic Drivers, Tumor Microenvironment and Resistance to Therapy

**DOI:** 10.3390/ijms21197139

**Published:** 2020-09-27

**Authors:** Elena Shklovskaya, Helen Rizos

**Affiliations:** Department of Biomedical Sciences, Faculty of Medicine, Health and Human Sciences, Macquarie University, Sydney 2109, Australia; helen.rizos@mq.edu.au

**Keywords:** immune checkpoint blockade, immunotherapy response biomarker, PD-L1 immune checkpoint, PD-L1 regulation, tumor microenvironment

## Abstract

Immunotherapies blocking immune inhibitory receptors programmed cell death-1 (PD-1) and cytotoxic T-lymphocyte-associated protein-4 (CTLA-4) on T-cells have dramatically improved patient outcomes in a range of advanced cancers. However, the lack of response, and the development of resistance remain major obstacles to long-term improvements in patient outcomes. There is significant interest in the clinical use of biomarkers to improve patient selection, and the expression of PD-1 ligand 1 (PD-L1) is often reported as a potential biomarker of response. However, accumulating evidence suggests that the predictive value of PD-L1 expression in tumor biopsies is relatively low due, in part, to its complex biology. In this review, we discuss the biological consequences of PD-L1 expression by various cell types within the tumor microenvironment, and the complex mechanisms that regulate PD-L1 expression at the genomic, transcriptomic and proteomic levels.

## 1. Introduction

Therapeutic antibodies that block the programmed cell death-1 (PD-1) immune inhibitory receptor or its ligand PD-L1 (CD274, B7-H1) have produced remarkable improvements in many patients with advanced malignancies. The clinical efficacy of PD-1 blockade was initially demonstrated in melanoma, renal, bladder and lung cancers and Hodgkin’s disease [1,2,3,4,5]. Currently, several therapeutic antibodies blocking either PD-1 (nivolumab, pembrolizumab, cemiplimab) or PD-L1 (atezolizumab, durvalumab, avelumab) have been FDA-approved for multiple indications, and additional antibodies directed at these inhibitory checkpoints have entered clinical trials.

PD-L1 expression in tumor biopsies has emerged as an important biomarker of response to immune checkpoint blockade directed at the PD-1 axis. Four diagnostic companion PD-L1 immunohistochemistry tests have been approved for use with PD-1/PD-L1 blocking antibodies in selected cancer subtypes (Dako platform 22C3, 28-8 and Ventana platform SP263, SP142) [6]. Despite a good concordance between these tests [6], PD-L1 expression remains an imperfect biomarker of response to PD-1 blocking immunotherapies, as many patients with PD-L1 expression in the tumor do not respond to treatment while a subset of patients with PD-L1-negative tumors will respond to PD-1 blockade [7,8]. These observations emphasize the need to better understand PD-L1 biology. Several recent studies have provided new insights into the complexity of PD-1/PD-L1 regulatory networks within the tumor microenvironment. In this review, we will discuss some of these recent findings in the context of PD-L1 expression within the tumor microenvironment, and changes in PD-L1 expression during tumor evolution.

## 2. The Significance of PD-L1 Expression on Tumor and Non-Tumor Cells

A classification based on tumor PD-L1 expression and the presence of tumor-infiltrating T-cells (TILs) has initially been developed for melanoma, to help select patients likely to respond to immunotherapies [7,9,10,11] (Figure 1). Melanoma tumors can be classified as (i) PD-L1-positive, TIL-positive (“hot” tumors, 35%), (ii) PD-L1-negative, TIL-negative (“cold” tumors, 40%), (iii) PD-L1-negative, TIL-positive (20%) or (iv) PD-L1-positive, TIL-negative (5%) tumors [11,12]. These subgroups are not consistently defined across tumor types however, and do not provide the predictive accuracy necessary for routine clinical management. In addition, different methods of staining and scoring, such as the use of different positivity thresholds, have contributed to variable conclusions regarding the role of tumor PD-L1 expression as a biomarker predictive of response to anti-PD-1-based therapies.

PD-L1 expression on non-cancer cells has also been shown to have predictive value in several cancers [13]. For instance, high expression of PD-L1 on immune cells infiltrating the tumor was shown to be an independent predictor of better melanoma patient outcomes following tumor resection [14], and a better predictor of response to PD-1 blockade than PD-L1 expression on tumor cells in immunogenic cancers such as melanoma, non-small cell lung cancer and urothelial cancer [3,4,15]. Expression of PD-L1 on non-tumoral cells in the tumor microenvironment can occur in the absence or presence of PD-L1 expression on tumor cells, and expression on both types can play a role in immune-mediated tumor control. Elegant animal studies by Arlene Sharpe’s group demonstrated that both tumor- and host-derived PD-L1 can restrict anti-tumor immunity and their relative contributions may relate to the level of tumor immunogenicity. For instance, PD-L1 expression on immunogenic MC38 colorectal tumor cells directly suppressed CD8 T-cell cytotoxicity and was dominant in suppressing anti-tumor immunity [16]. In contrast, PD-L1 expression on non-tumor cells was required for the immune evasion of the poorly immunogenic *Braf/Pten*-mutant melanoma [16]. Importantly, differences in the expression level of PD-L1 within the tumor microenvironment did not account for the distinct contributions of tumor- versus host-derived PD-L1 in modulating anti-tumor immune responses [16].

## 3. PD-L1 Expression on Non-Tumor Cells

Within the solid tumor microenvironment, PD-L1 can be expressed on many cells of hematopoietic origin, often collectively referred to as “immune infiltrate”, including dendritic cells [17], tumor-associated macrophages (TAMs) [4,18], myeloid-derived suppressor cells (MDSCs) [19] and T-cells [3]. PD-L1 can also be expressed on the non-hematopoietic stromal elements, principally the endothelial cells of the tumor vasculature [20] and cancer-associated fibroblasts [21]. Expression of PD-L1 by these diverse stromal elements has both overlapping and non-redundant roles in immune-mediated tumor control, particularly in the context of therapies targeting the PD-1/PD-L1 inhibitory axis [3,4]. 

*Dendritic cells (DCs).* Tumor-associated DCs upregulate PD-L1 mainly in response to interferon-γ (IFNγ) released by tumor-infiltrating T-cells. Since activated T-cells in the tumor bed represent the principal local source of IFNγ, PD-L1 expression by DCs and other myeloid cells can be regarded as a surrogate of T-cell activation resulting from tumor antigen recognition [22,23]. Type I IFNs can also upregulate PD-L1 on myeloid cells, augmenting cytotoxic T-cell responses via improved antigen presentation, thus enhancing the likelihood of clinical response to PD-1 blockade [24]. 

PD-L1 expressed on DCs provides a direct T-cell inhibitory input via PD-1 but also helps override T-cell activation in the context of antigen recognition [17,25]. PD-L1 has two binding partners, the inhibitory receptor PD-1 on T-cells and the co-stimulatory molecule CD80 (B7.1) on antigen-presenting cells. In the tumor microenvironment, DCs express both PD-L1 and CD80, with the amount of PD-L1 greatly exceeding that of CD80. During the DC-T-cell cross-talk, PD-L1 on the DC binds to and sequesters CD80 in *cis*, while the excess of unbound PD-L1 interacts with PD-1 on the T-cell, resulting in functional inactivation. Anti-PD-L1 antibodies release CD80 from this sequestered form and re-instate the CD80-CD28 co-stimulatory interaction while simultaneously blocking the PD-L1-PD-1 inhibitory pathway, resulting in augmented T-cell activation upon antigen recognition [17] (Figure 2). Of note, the inability of PD-L1 to bind CD80 in *cis* resulted in attenuated immune responses, including anti-tumor responses [25].

*T-cells.* Tumor antigen-specific T-cells within the tumor microenvironment often express multiple inhibitory receptors including PD-1 and this expression profile is indicative of T-cell inactivation, also termed exhaustion or dysfunction [26]. However, T-cells also express PD-L1, which is rapidly upregulated following T-cell activation and is important for T-cell survival [27]. PD-L1-deficient T-cells are more susceptible to killing by cytotoxic T-cells, indicating that PD-L1 protects T-cells undergoing clonal expansion and supports optimal protective immunity [27]. PD-L1-deficient T-cells exhibit enhanced rates of apoptosis, reduced metabolism, diminished production of inflammatory cytokines and abnormal expression of tissue-homing receptors both at baseline and after activation [28].

The ligation of PD-L1 expressed by T-cells can promote tumor immune escape via diverse mechanisms [29]. First, T-cell-expressed PD-L1 can engage with PD-1 expressed on macrophages to promote M2 polarization. Second, PD-L1 on T-cells can engage with PD-1 expressed on other T-cells to reduce production of effector cytokines IFNγ and tumor necrosis factor (TNF)α. Third, T-cell-expressed PD-L1 can function as a receptor in T-cells. This so-called “back-signaling” can promote T-helper 1 (Th1)-to-Th17 switch in CD4 T-cells [29], a non-responsive (anergic) phenotype in CD8 T-cells [29] and apoptosis in activated T-cells [30]; the ligation of PD-L1 on T-cells was as efficient as PD-1 ligation in suppressing T-cell functionality [29]. In addition to PD-1 and PD-L1, activated T-cells can also express CD80 known to restrain T-cell effector function through CTLA-4 [31]; the role for PD-L1–CD80 interactions in T-cell bidirectional signaling remains to be addressed. In summary, T-cell-expressed PD-L1 contributes to the accumulation of dysfunctional T-cells in the tumor, via enhanced clonal survival coupled with reduced effector functions. Targeting T-cell-expressed PD-L1 offers new therapeutic opportunities, in particular for T-cell-infiltrated cancers with low/absent PD-L1 expression on tumor cells. 

*Tumor-associated macrophages (TAMs).* Generally, a high density of myeloid cells in the tumor correlates with reduced T-cell infiltration and a poor prognosis (reviewed in [32]). Myeloid precursors are recruited to tumors in a T-cell-independent manner, and the regulation of PD-L1 expression in myeloid cells, particularly in the immunologically “cold” tumors, does not necessarily reflect a concurrent T-cell response [33]. Hypoxia and the accompanying angiogenesis-promoting angiopoietins recruit monocytes expressing the cognate TIE2 receptor to tumors, where these myeloid precursors preferentially migrate to the hypoxic areas and differentiate into TAMs [34,35,36]. Accordingly, hypoxia is an important regulator of TAM biology [37]. PD-L1 expression in myeloid cells is upregulated directly by the hypoxia-inducible factor (HIF)-1α that binds to the PD-L1 promoter to induce PD-L1 transcription [38]. Furthermore, TAMs upregulate PD-L1 expression by assuming aerobic glycolysis [39] while also secreting TNFα that promotes aerobic glycolysis in cancer cells [40] and augments PD-L1 expression on myeloid cells [41]. Sustained chronic inflammation enhances PD-L1 expression on myeloid cells through the cyclooxygenase 2/prostaglandin E2 pathway [19] and interleukin (IL)-6 production [42]. Type I and type II interferons augment PD-L1 expression on myeloid cells in the context of immunotherapies that activate innate or adaptive anti-cancer immune responses, respectively [43,44,45].

A recent discovery of macrophages expressing the PD-1 receptor [46] adds a layer of complexity to the biology of the PD-1/PD-L1 axis in tumor myeloid cells. PD-1-expressing TAMs were exclusively of the pro-tumorigenic M2-polarized type [46], which is in agreement with Diskin et al., who reported that T-cell-expressed PD-L1 could engage with TAM-expressed PD-1 to promote M2 polarization driving a cancer-permissive environment [29]. It is unclear to what extent anti-PD-1 therapeutic antibodies directly modify the biology of PD-1-expressing TAMs, but myeloid cell-targeting therapies, especially a blockade of the colony stimulating factor 1 (CSF1) receptor signaling, potently synergize with immunotherapy in preclinical models of cancer. Several ongoing clinical trials address the safety and efficacy of myeloid cell-targeting drugs combined with immune checkpoint inhibitors in advanced melanoma, renal and lung cancers (reviewed in [47]).

*Endothelial cells.* Abnormal angiogenesis is one of the hallmarks of cancer. Vascular endothelial cells closely guard immune cell extravasation and accumulation of T-cells in the tumor, by regulating T-cell adhesion and modulating functions of T-cells that transit through the vessel wall [48,49] (Figure 2). Endothelial cell-expressed PD-L1 suppressed CD8 T-cell cytotoxicity and cytokine production without affecting T-cell activation [50,51], and enhanced the inhibitory function of regulatory T-cells in vitro [52]. Tumor endothelial cells could also kill transmigrating CD8 T-cells via the Fas ligand–Fas interactions, while sparing regulatory T-cells that are relatively resistant to Fas-mediated apoptosis [49]. In turn, T-cells modulate endothelial cell functions via the local production of IFNγ and TNFα [20,50,51,53].

Anti-angiogenic therapies trigger vascular regression and/or blood vessel normalization, allowing for the extravasation of cytotoxic T-cells, release of IFNγ and TNFα and the resulting increase in PD-L1 expression on endothelial cells [20,53,54]. This effect could be replicated in vitro, in endothelial cells exposed to IFNγ and TNFα [50,51]. IFNγ and TNFα also triggered the production of immunosuppressive cytokines IL-6 and transforming growth factor (TGF)β that further enhanced PD-L1 expression on endothelial cells (reviewed in [55]).

Endothelial cell expression of PD-L1 plays a role in resistance to anti-angiogenic therapy, as anti-angiogenic therapy synergized with PD-1/PD-L1 blockade in several preclinical models of cancer [20,53]. Furthermore, anti-angiogenic therapy combined with anti-PD-1 facilitated the formation in the tumor tissue of highly specialized capillaries known as high endothelial venules (HEVs), structures normally present in lymph nodes where they serve as dedicated sites of T-cell homing, likely indicative of the development of tertiary lymphoid structures within the tumor [20]. Thus, PD-L1 expression by the tumor vasculature plays a role in limiting access and restricting the function of T-cells that enter the tumor tissue.

In summary, the non-overlapping expression patterns and the non-redundant functions of PD-1 and PD-L1 expressed on multiple cell types within the tumor microenvironment strongly suggest a potential benefit for combined PD-1/PD-L1 targeting. Combined PD-1/PD-L1 blockade is currently being investigated for safety and tolerability (clinicaltrials.gov identifier: NCT02118337).

## 4. Expression of PD-L1 by Tumor Cells

Mechanisms responsible for PD-L1 expression by tumor cells can be divided broadly into constitutive and inflammation-driven expression.

**Constitutive expression of PD-L1 on tumor cells.** Tumor cell-intrinsic (constitutive) expression of PD-L1 is not linked to the ongoing immune response and can be observed in the absence or presence of T-cells in tumor biopsies. Multiple mechanisms can contribute to tumor cell-intrinsic PD-L1 expression, including genetic, post-transcriptional and post-translational regulation.

*Genetic events that determine PD-L1 expression*. The *PD-L1*/*CD274* gene is located on chromosome 9p24.1 in proximity to the second PD-1 ligand PD-L2 (*PDCDLG2*) and the gene encoding Janus kinase 2 (*JAK2*), the downstream kinase involved in IFNγ receptor signaling. Copy number amplifications of 9p24.1 have been associated with increased PD-L1 expression in several cancers, and occur most commonly in mediastinal large B-cell lymphoma, classical Hodgkin’s lymphoma [56] and triple-negative breast cancer [57], but have also been described in ovarian, head and neck, bladder, cervical cancers, sarcomas and colorectal cancers, albeit at lower frequencies [8,58]. Enhanced signaling via JAK2 in cancers with 9p24.1 amplifications, contributed to mixed inflammatory and constitutive tumor-derived PD-L1 expression [56]. In addition, tumors with *CD274* genetic gains demonstrated higher mutational loads compared to non-amplified cases and responded particularly well to PD-1 blockade [5,59]. Deletions of PD-L1 are also commonly found in melanoma and non-small cell lung cancer, and represent one of the mechanisms leading to the lack of tumor cell PD-L1 expression [58,60].

*Aberrant oncogenic signaling.* Oncogenic signaling is an important regulator of tumor PD-L1 expression (Figure 3). Several oncogenic transcription factors including MYC, RAS and STAT3 individually or co-operatively promote PD-L1 expression in tumor cells.

Activation and overexpression of MYC is observed in many cancers and is directly linked to tumorigenesis. MYC binds the promoter region of PD-L1 directly, upregulating PD-L1 expression in T-cell leukemia, hepatocellular carcinoma, melanoma and colorectal cancer [61]. In non-small cell lung cancer, a significant positive correlation was observed between MYC and PD-L1 expression by immunohistochemistry [62].

Oncogenic RAS signaling increases PD-L1 expression though c-Jun binding, such as in BRAF inhibitor-resistant melanoma [63], and via stabilization of PD-L1 mRNA, such as in lung and colorectal tumors [64]. Furthermore, inactivation of the tumor suppressor TP53, which is often associated with activating RAS mutations in lung adenocarcinoma and increased mutational load due to DNA damage repair defects, drives T-cell activation and immune-mediated elevation of PD-L1 expression, translating into a strong clinical benefit of PD-1 blockade in such patients [65].

Activating mutations in epidermal growth factor receptor (EGFR) drive the strong constitutive expression of tumor PD-L1 in a subset of patients with non-small cell lung cancer (reviewed in [66]). Expression of mutant EGFR was sufficient to induce PD-L1 expression in bronchial epithelial cells, while EGFR targeting reduced PD-L1 expression in EGFR-mutant tumor cells [67]. It appears that EGFR promotes PD-L1 expression by directly activating PD-L1 gene expression via the MEK/ERK/c-Jun pathway [68], as well as indirectly via IL-6/JAK/STAT3 signaling [69]. Similarly, activating mutations in the fibroblast growth factor receptor 2 (FGFR2) in colorectal cancer activate PD-L1 expression via the JAK/STAT3 pathway [70].

Oncogenic activation of STAT3 drives strong PD-L1 expression in lymphoma, by increasing STAT3 transcriptional activity and ensuring robust binding to the PD-L1 promoter [71]. Nucleophosmin-anaplastic lymphoma kinase fusion protein (NPM/ALK)-carrying T-cell lymphomas also strongly express PD-L1 via NPM/ALK-activated STAT3 [72].

In de-differentiated cancers with features of epithelial-to-mesenchymal transition (EMT), additional signaling pathways maintain PD-L1 expression, including Yes-associated protein (YAP) and β-catenin pathways (Figure 3). The Hippo pathway effector and transcriptional co-activator YAP acts in concert with the TEA domain (TEAD) family of transcription factors to regulate PD-L1 expression in de-differentiated cancers. The PD-L1 promoter has two putative TEAD binding sites [73], and YAP recruits TEAD transcription factors to the PD-L1 promoter region [73,74]. YAP augments PD-L1 expression in EGFR inhibitor-resistant lung cancer [73,74], pancreatic cancer [75], mesothelioma [76] and BRAF inhibitor-resistant melanoma [77]. The inhibition of YAP with a small molecular inhibitor or via gene knockdown decreased PD-L1 mRNA and protein expression in mesothelioma cells [76]. In human non-small cell lung cancer, nuclear YAP staining on immunohistochemistry was associated with PD-L1 expression [73,76].

Activation of β-catenin signaling contributes to PD-L1 expression in de-differentiated cancers. Mechanistically, loss of the epithelial marker E-cadherin frees the E-cadherin-associated β-catenin for cytoplasmic translocation, GSK-3β-mediated ubiquitination and proteosomal degradation [78]. Activation of wingless (WNT) signaling blocks the GSK-3β-containing destruction complex, allowing for β-catenin translocation to the nucleus and association with the TCF/Lef-1 family of transcription factors that bind the PD-L1 promoter to upregulate PD-L1 expression (Figure 3). β-catenin signaling also activates Zinc finger E-box binding homeobox 1 (ZEB-1) transcription factor, one of the major mediators of EMT. ZEB1 augments PD-L1 gene expression either directly by binding to the PD-L1 promoter [79] or indirectly, by repressing microRNA (miR)-200 that regulates PD-L1 mRNA decay [80]. The ZEB1-miR-200 axis is one of the major regulators of PD-L1 expression in EMT.

Epigenetic and post-transcriptional mechanisms also contribute to the enhanced PD-L1 expression in EMT states. For example, demethylation of the PD-L1 promoter via TGFβ1-dependent repression of the DNA methyltransferase DNMT1 allowed for expression of PD-L1 upon TNFα-mediated NF-kB activation [81]. Importantly, tumor PD-L1 expression in EMT states is associated with poor responses to PD-1 blockade in a subset of patients [80,82,83] due to tumor expression of additional inhibitory ligands [84,85,86,87], loss of differentiation antigens by tumor cells [88] and activation of T-cell exclusion mechanisms [89,90,91], collectively contributing to reduced T-cell infiltration and tumor immune escape [86,92,93].

Finally, the AKT-mTOR pathway serves as a convergence point for the activation of many oncogenic pathways, and is involved in regulating PD-L1 expression at the protein level via the regulation of protein synthesis and lysosomal protein degradation [94]. Accordingly, the mTOR inhibitor rapamycin significantly reduced tumor burden in mice bearing carcinogen-induced lung tumors [94].

In addition to aberrant oncogenic signaling, other genetic mechanisms regulate PD-L1 expression. The 3′ untranslated region (3′-UTR) of the *PD-L1* gene is commonly disrupted by structural alterations leading the production of multiple stable aberrant transcripts and the resultant elevation of PD-L1 expression [95]. An increase in PD-L1 expression due to 3′-UTR disruption is thought to interfere with post-transcriptional regulation such as miR-mediated control, resulting in a decreased mRNA decay rate [80,95,96,97,98]. A super-enhancer located between the PD-L1 and PD-L2 genes has recently been identified [99] and shown to maintain expression of PD-L1 independently of IFNγ. It is subject to epigenetic regulation and sensitized cancer cells to PD-1 blockade [99].

### Inducible Expression of PD-L1

*Transcriptional regulation of PD-L1 expression.* A multitude of cytokines and growth factors can initiate or augment PD-L1 expression on cancer cells and in the tumor microenvironment. Of these, IFNγ produced by TILs as a result of T-cell recognition of tumor antigens is the most potent inducer of non-constitutive PD-L1 expression in cancer cells, and in the tumor microenvironment (reviewed in [100]). Loss of IFNγ responsiveness in tumor cells may result from inactivating mutations in JAK1/2 disrupting INFγ signaling [101], leading to tumor immune escape and disease progression. Mechanistically, IFNγ acts via the JAK/STAT/IRF axis [102] and NF-kB [103] to stimulate PD-L1 production. Tumor cells with disrupted IFNγ signaling had a defective PD-L1 upregulation and were efficiently controlled by the immune system in a mouse model of melanoma [104]. Such IFNγ signaling-deficient cancer cells required cooperation from IFNγ signaling-sufficient and PD-L1-positive cells for immune escape, indicating the importance of PD-L1 and environmental clues for tumor immune escape. These data also suggest that disrupted IFNγ signaling is advantageous for tumors in the context of immunotherapy [101,104].

TNFα can induce PD-L1 expression in the absence of IFNγ [41]. However, in T-cell-infiltrated tumors, both cytokines are often co-produced by T-cells, and an increase in PD-L1 expression at the height of an immune response is likely to result from the concerted action of TNFα and IFNγ synergizing to increase PD-L1 production. Other inflammatory mediators that upregulate PD-L1 expression in tumor and non-tumor cells include IL-6 [42], prostaglandin E2 [19] and HIF-1α [38].

At the epigenetic level, regulation of PD-L1 expression is most intricately linked to DNA methylation. Hypermethylation of the PD-L1 promoter prevented PD-L1 expression [12], while a constitutive expression of PD-L1 in melanoma biopsies and cell lines was associated with global DNA hypomethylation patterns [105,106]. Interestingly, some of the hypomethylated cancer cells also produced high concentrations of inflammatory cytokines including IFNγ, IL-6 and IL-8 [107], suggesting that constitutive PD-L1 expression in epigenetically dysregulated cancers could be maintained, at least in part, by the IFNγ and/or IL-6-dependent feedback loop. Other epigenetic mechanisms such as histone deacetylation and aberrant expression of the Enhancer of zeste homolog 2 (EZH2) contribute to the control of PD-L1 expression in cancer cells by limiting transcription factor access to the PD-L1 promoter region (reviewed in [12]). Accordingly, targeting epigenetic mechanisms responsible for the low/absent PD-L1 expression on cancer cells has emerged as an attractive strategy to improve efficacy of PD-1-directed immunotherapies in selected patient cohorts ([12]). 

*Post-transcriptional regulation of PD-L1 expression*. Multiple miRs regulate PD-L1 expression either at baseline or in response to IFNγ, by mediating mRNA degradation or by inhibiting translation. A number of miRs can bind to the 3′UTR of the PD-L1 gene and have been shown to directly suppress PD-L1 expression, including miR-513 [96], miR-155 [108], miR-17-5p [109], miR-33a [110], miR-34a [98] and multiple others (Table 1). In contrast, only a few miRs have been shown to increase PD-L1 expression by activation of PD-L1 transcription inducers. For example, miR-18a enhanced PD-L1 expression in cervical cancer by targeting the tumor suppressor PTEN and the Wnt/β-catenin pathway inhibitor SOX6 to activate the PI3K/AKT, MEK/ERK and Wnt/β-catenin pathways, respectively [111]. miR-20b, -21 and -130b upregulated PD-L1 expression in colorectal cancer also by targeting PTEN [112], while miR-135 augmented PD-L1 expression in lung cancer by suppressing the ubiquitination of proteins in the JAK/STAT signaling pathway [113].

## 5. Regulation PD-L1 Protein Expression

*Glycosylation.* PD-L1 glycosylation regulates PD-L1 protein stability and degradation. Several key regulators of glycosylation have been identified and these may offer new options for therapeutic PD-1 blockade [42,141]. Nonglycosylated PD-L1 is an unstable protein that is rapidly degraded by the ubiquitin/proteasome system after being phosphorylated by glycogen synthase kinase 3β (GSK3β) [142]. PD-L1 is heavily glycosylated [142], as are other immune inhibitory receptors and ligands including PD-1 [143]. Glycosylation improves PD-L1 stability [144] but also creates a therapeutic vulnerability, where abnormal glycosylation such as that triggered by a commonly used anti-diabetic drug metformin results in endoplasmic reticulum-associated degradation of PD-L1 and improved immune responses [145].

Glycosylation is important for the PD-L1 interaction with PD-1, and antibodies specifically targeting a glycosylated form of PD-L1 triggered immune-mediated rejection of mouse tumors expressing the human version of PD-L1 [143]. A similar result has recently been achieved with a therapeutic antibody directed against the glycosylated form of PD-1, which was superior to both nivolumab and pembrolizumab in a humanized animal model of triple-negative breast cancer [146]. These studies indicate that variations in PD-1 and PD-L1 glycosylation patterns may alter therapeutic antibody binding and therefore skew immunotherapy outcomes, particularly for PD-L1 blockade.

Aberrant glycosylation of PD-L1 in cancer cells may provide some insight into the validity of PD-L1 detection as a predictive biomarker of immunotherapy response. Morales-Betanzos et al. demonstrated, using mass spectrometry and immunohistochemistry on 22 human melanoma samples, a discrepancy in PD-L1 expression assessed by the two methods, which the authors attributed to the aberrant PD-L1 glycosylation interfering with the antibody binding on immunohistochemistry [147]. Thus, at least in some of the “PD-L1-negative” tumors, glycosylation variants of PD-L1 may be expressed that are not detectable by immunohistochemistry.

*Polyubiquitination and degradation of PD-L1.* PD-L1 degradation is regulated by ubiquitination/proteasome and autophagy pathways [148,149,150,151], and targeting PD-L1 degradation has emerged as an alternative strategy to improve immunotherapy efficacy. COP9 signalosome 5 (CSN5) was required for TNFα-mediated PD-L1 stabilization in cancer cells, by inhibiting the ubiquitination and degradation of PD-L1 [148]. Blockade of CSN5 by curcumin destabilized PD-L1 and sensitized cancer to immunotherapy [148]. Similarly, activation of autophagy by verteporfin reduced PD-L1 expression in cancer cells [152].

The transmembrane protein CMTM6 and the closely related CMTM4 regulate PD-L1 cell surface expression by associating with PD-L1 in recycling endosomes and protecting PD-L1 from ubiquitination (by the E3 ubiquitin ligase STUB1) and the resulting proteosomal degradation [149,150]. Cyclin D-CDK4 kinase also promotes PD-L1 ubiquitination and degradation via the cullin 3 - SPOP E3 ligase (SPOP is stabilized by Cyclin D-CDK4-mediated phosphorylation) [151]. Thus, CDK4/6 inhibitors synergized with PD-1 blockade in mouse tumor models [151]. Taken together, these observations highlight the multilayered regulation of PD-L1 expression, from transcription to protein degradation. Although most of these mechanisms have been studied in cancer cell lines, many may also operate in non-cancer cells. The exact contribution of these mechanisms to the expression of PD-L1 in myeloid and immune cells is yet to be defined.

*Soluble PD-L1*. Soluble PD-L1 (sPD-L1) is frequently detected in the blood of cancer patients. Three mechanisms can contribute to the release of PD-L1 into circulation: alternatively spliced transcripts [153,154], release of PD-L1 associated with extracellular vesicles such as exosomes [155,156] and proteolytical cleavage from the surface of PD-L1-expressing cancer- and non-cancer cells [157].

In several studies, high levels of circulating sPD-L1 were associated with poor patient outcomes in melanoma, lung and gastric cancers and lymphoma [155,158,159,160] and with reduced response to PD-1 blockade in melanoma [161]. Of note, sPD-1 was also found in the blood of melanoma patients and, together with sPD-L1, predicted poor response to PD-1 blockade [161]. Some of sPD-L1 retains biological activity, including exosome-associated PD-L1 [155,156] and a truncated isoform that retains the ability to form dimers and interact with PD-1 [154]. Biologically active sPD-1 and sPD-L1 may differentially interfere with therapeutic antibodies directed against PD-L1 and PD-1, respectively.

## 6. Temporal Changes in PD-L1 Expression

*Early increase in PD-L1 expression is associated with response to treatment.* Analysis of longitudinal biopsies in patients treated with PD-1 inhibitors revealed that an increase in PD-L1 expression compared to baseline in the first eight weeks of therapy correlated with an increase in tumor-infiltrating T-cells and was an indicator of a treatment response [162,163]. This increase in PD-L1 expression was driven by T-cell-derived IFNγ [164] and observed on both cancer and non-cancer cells [163], with macrophage PD-L1 expression often protracted compared to cancer cells [164]. A corresponding T-cell clonal expansion was detectable in blood within one–three weeks of treatment [165], indicative of clonal replacement [166] in addition to the “invigoration” of the existing dysfunctional T-cells [165]. A similar increase in PD-L1 expression in longitudinal biopsies, combined with an immune gene expression signature, was observed in patients treated with anti-CTLA-4 antibodies [167]. Thus, an increase in PD-L1 expression in longitudinal tumor biopsies, on tumor and non-tumor cells, was a strong indicator of immunotherapy response, not limited to anti-PD-1 treatment.

*PD-L1 expression in de-differentiated tumors is associated with treatment resistance.* Tumors from patients who progress while on targeted therapy or immunotherapy often display features associated with hypoxia, angiogenesis, EMT program and de-differentiation [88,168,169]. These features are associated with poor patient responses to PD-1 blockade in melanoma, lung, head and neck, breast cancer and other cancer types, despite PD-L1 expression in tumor lesions [80,81,82,83,84,130]. TGFβ produced by tumor stromal cells helps establish the EMT program in cancer cells by driving the expression of key transcription factors ZEB, Twist, Snail and Slug [170,171,172] while contributing to T-cell exclusion [89]. Signaling via HIF-1α [38], β-catenin [173], AXL tyrosine kinase [174,175], YAP [73,74] and ZEB1 pathways [79,80] variably contributes to augmented PD-L1 expression in de-differentiated tumors. Accordingly, pharmacological inhibition of these pathways decreased PD-L1 expression in the corresponding in vitro and in vivo models of cancer [173,174,175] and conferred sensitivity to PD-1/PD-L1 axis blockade [171]. TGFβ in particular represents an attractive therapeutic target for potentially reversing EMT. However, inhibition of TGFβ signaling released matrix metalloproteinase-9 by stromal fibroblasts, resulting in cleavage of PD-L1 from the surface of PD-L1-expressing cancer cells and myeloid cells and generation of soluble PD-L1, thus potentially desensitizing to anti-PD-1 treatment [157]. Tumor heterogeneity in EMT states also contributes to variable inter- and intralesional PD-L1 expression [88], including PD-L1 expression on cancer stem cells (via increased glycosylation resulting in improved PD-L1 protein stability), further contributing to tumor immune escape [141].

## 7. Concluding Remarks

Immunotherapies targeting the PD-1/PD-L1 axis have demonstrated a remarkable efficacy in a range of cancers. However, many patients fail to respond, and treatment resistance remains a major obstacle with only a small proportion of patients experiencing complete responses. Expression of PD-L1 in tumor biopsies, while an imperfect biomarker predictive of response to PD-1/PD-L1-based therapies, is constantly re-evaluated to improve its predictive value. Advances in immunohistochemistry staining and scoring of PD-L1 expression in tumor lesions, independent assessment of PD-L1 expression on tumor and non-tumor cells, combining PD-L1 expression scores with other parameters such as T-cell infiltration and T-cell proximity to tumor cells, tumor mutational load and signatures of IFNγ response help improve the predictive value of PD-L1 expression in tumor biopsies [176,177,178]. Yet, new evidence has uncovered some unexpected twists in the PD-1/PD-L1 biology that need to be considered when analyzing the potential outcomes of treatment or predictive value of PD-L1 expression in certain patient cohorts. A protective role of PD-L1 expressed on T-cells [27], the discovery of PD-L1 bidirectional signaling in T-cells [29] and PD-L1-mediated sequestration of T-cell activation signals on dendritic cells [17] indicate non-redundant roles for PD-1 and PD-L1 blockade and the potential for combination therapies. PD-1 expression on macrophages [46] and tumor cells [179] identifies these cell types as direct targets of PD-1 blockade and indicates the potential for unexpected treatment outcomes. For PD-1-expressing T-cell subsets, shifting the balance of response from PD-1^+^CD8^+^ T-cells towards PD-1^+^ regulatory T-cell activation may reinforce immunosuppression and promote adaptive immune resistance [180]. Identification of sPD-L1 isoforms retaining biological activity [154] suggests the possibility of therapeutic anti-PD1 neutralization and immunotherapy attenuation. Characterization of glycosylated PD-L1 isoforms [147] indicates the potential for interference with immunohistochemistry-based PD-L1 detection and “false negative” results. Finally, a better understanding of the mechanisms driving antigen loss and T-cell exclusion in tumors with EMT features, as well as the strong association with progression on single-agent anti-PD-1, suggest a rationale for first-line combination immunotherapies in such patients.

There are also some exciting new treatments under investigation. Therapeutic targeting of glycosylated PD-L1 [143] and PD-1 isoforms [146], and enhancing PD-L1 degradation by disrupting endosomal recycling [149,150] offer new ways of targeting the PD-1/PD-L1 pathway. Epigenetic targeting of tumor PD-L1 expression [12] can also improve tumor-specific antigen expression and the resulting T-cell recognition, thus creating new therapeutic vulnerabilities. Finally, a partial reversal of tumor EMT with TGFβ-, AXL- and YAP-targeting therapies [76,171,175] may re-sensitize tumors to immunotherapies by restoring antigen expression and allowing tumor T-cell infiltration.

## Figures and Tables

**Figure 1 ijms-21-07139-f001:**
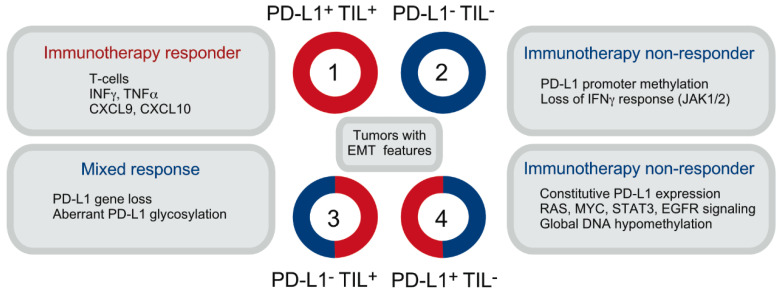
Classification of melanoma tumors based on the degree of T-cell infiltration (TIL) and PD-L1 expression. Categories are aligned with the dominant patient response and mechanisms responsible for high or low/absent PD-L1 expression. Note that tumors with epithelial-to-mesenchymal transition (EMT) features can be found in any of the four groups.

**Figure 2 ijms-21-07139-f002:**
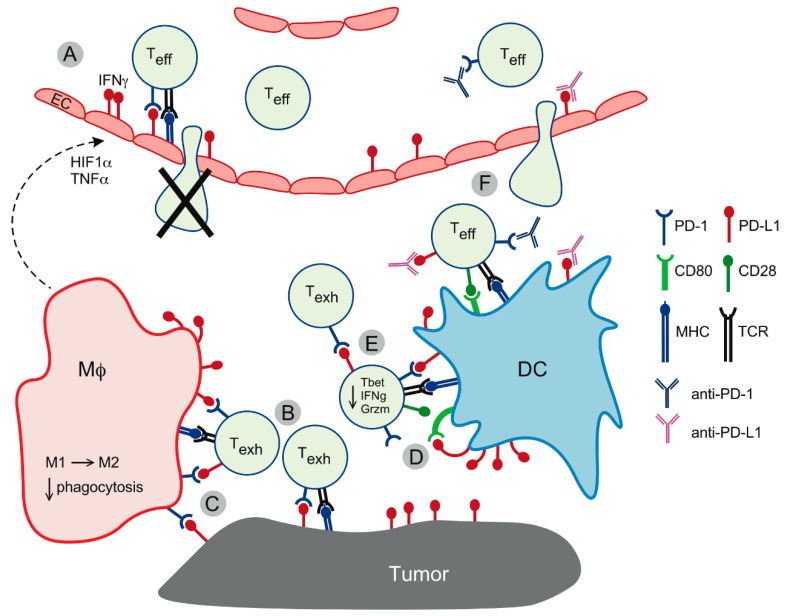
PD-L1-mediated cellular interactions in the tumor microenvironment. (**A**) PD-L1 upregulation on blood vessel endothelial cells (EC) in response to T-cell-derived IFNγ and macrophage-derived hypoxia-inducible factor 1α (HIF1α) and tumor necrosis factor α (TNFα) functionally inactivates T-cells and reduces their transmigration into the tumor bed. Endothelial cells can also induce Fas-dependent T-cell death in migrating T-cells. (**B**) PD-L1 interacts with PD-1 on T-cells maintaining a state of exhaustion/dysfunction (T_exh_). (**C**) PD-L1 expressed on T-cells interacts with PD-1-positive macrophages (Mϕ), promoting M2 polarization and functional impairment. (**D**) PD-L1 on dendritic cells (DC) sequesters CD80 in *cis*, preventing it from interacting with CD28 on T-cells and thus abolishing T-cell activation. Excess of PD-L1 binds PD-1, contributing to T-cell exhaustion. (**E**) Reverse signaling via PD-L1 on T-cells impairs effector functions, such as cytokine production and killing capacity, while at the same time protecting T-cells from death, thus contributing to the expansion of functionally impaired T-cell clones. (**F**) Therapeutic antibodies restore T-cell effector function (Teff) by blocking PD-1 and/or PD-L1 signaling to the T-cell and releasing PD-L1-bound CD80 for interaction with CD28, thus enhancing T-cell stimulation upon antigen recognition via the T-cell receptor (TCR). In addition, therapeutic antibodies improve T-cell recruitment to the tumor by blocking PD-L1 on endothelial cells.

**Figure 3 ijms-21-07139-f003:**
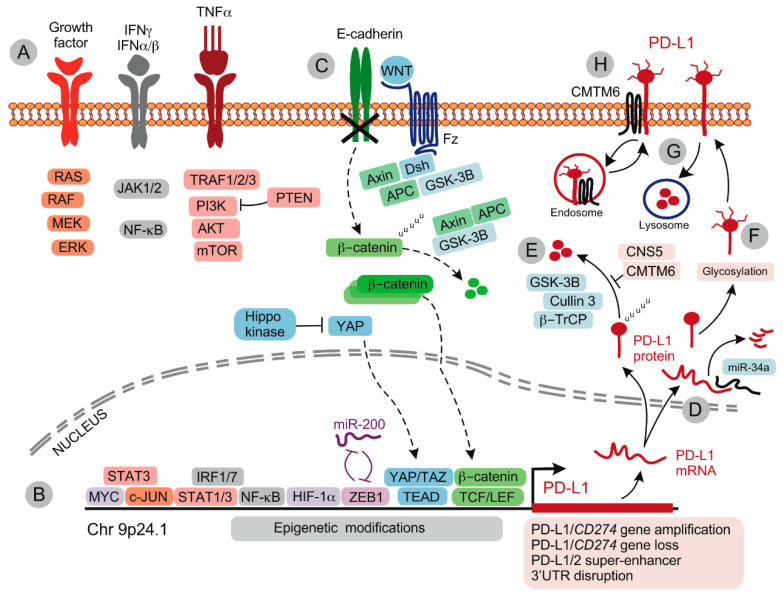
Regulation of PD-L1 expression in tumor cells. (**A**) Signaling via growth factor-, IFN- and TNFα receptors activates multiple signaling pathways that induce *PD-L1*/*CD274* gene expression. (**B**) Multiple transcription factors can induce PD-L1 expression. PD-L1 gene amplification or loss and epigenetic modifications modulate PD-L1 gene expression. (**C**) In de-differentiated cells undergoing epithelial-to-mesenchymal transition (EMT), loss of E-cadherin drives cytoplasmic translocation of β-catenin that is subject to ubiquitination/destruction by the Axin/APC/GSK-3B complex. Activation of Wnt/Fz/Dsh sequesters the β-catenin destruction complex and allows for the accumulation and nuclear translocation of β-catenin; β-catenin associates with TCF/LEF transcription factors to induce PD-L1 expression. ZEB1 (subject to control by miR-200) and YAP/TAZ-TEAD complexes maintain PD-L1 expression in de-differentiated phenotypes. (**D**) Upon transcription, PD-L1 mRNA is subject to regulation by miRs such as miR-34a. (**E**) PD-L1 protein is unstable, being rapidly ubiquitinated by GSK-3B and cullin 3/β-TrCP, and degraded. This process is antagonized by CNS5 and CMTM6. (**F**) Glycosylation increases PD-L1 protein stability. (**G**) Cell surface-expressed PD-L1 is internalized and undergoes lysosomal degradation. (**H**) Alternatively, interaction with CMTM6 stabilizes PD-L1 expression by supporting endosomal recycling and preventing proteolytic degradation. Abbreviations: APC, Adenomatous polyposis coli; β-TrCP, Beta-transducin repeats-containing protein; CMTM6, CKLF-like MARVEL transmembrane domain containing 6; CNS5, COP9 signalosome 5; Dsh, Dishevelled; Fz, Frizzled; GSK-3B, synthase kinase 3 beta; HIF-1α, Hypoxia-inducible factor-1α; NF-kB, Nuclear Factor kappa B; mTOR, mammalian target of rapamycin; MYC, Avian myelocytomatosis virus oncogene; PI3K, Phosphoinositide 3-kinase; PTEN, phosphatase and tensin homolog; STAT, signal transducer and activator of transcription; TCF/LEF, T-cell specific transcription factor/lymphoid enhancer binding factor; TEAD, TEA domain family member; TRAF, TNF receptor associated factor; WNT, wingless; YAP, Yes-associated protein; ZEB1, Zinc finger E-box-binding homeobox 1.

**Table 1 ijms-21-07139-t001:** MiR regulation of PD-L1 expression.

miR	Change in PD-L1 Expression	Cancer or Cell Model	Reference
miR-15a, 15b	Decrease	Mesothelioma	[114]
miR-16	Decrease	Mesothelioma Prostate cancer	[114] [115]
miR-17-5p	Decrease	Melanoma (BRAF inhibitor resistant)	[109]
miR-25	Decrease	Bone marrow stromal cells	[116]
miR-33a	Decrease	Lung cancer	[110]
miR-34a	Decrease	Acute myeloid leukemia Lung cancer B-cell lymphomas	[97,98] [117]
miR-93	Decrease	Bone marrow stromal cells	[116]
miR-106b	Decrease	Bone marrow stromal cells	[116]
miR-138-5p	Decrease	Colorectal cancer	[118]
miR-140	Decrease	Colorectal cancer Lung cancer	[119] [120]
miR-142-5p	Decrease	Pancreatic cancer Lung cancer (via PTEN)	[121] [122]
miR-148a-3p	Decrease	Colorectal cancer (MSI-high)	[123]
miR-152	Decrease	Gastric cancer	[124]
miR-155	Decrease	Endothelial cells (IFNγ/TNFα response)	[108]
miR-191-5p	Decrease	Colorectal cancer	[125]
miR-193a-3p	Decrease	Mesothelioma	[114]
miR-195	Decrease	Mesothelioma B-cell lymphomas Pancreatic cancer Prostate cancer	[114] [126] [127] [115]
miR-197	Decrease	Lung cancer (via CKS1B/STAT3)	[128]
miR-200 family	Decrease	Lung cancer Gastric cancer Hepatocellular carcinoma Breast cancer Acute myeloid leukemia	[80] [124] [129] [130] [131]
miR-217	Decrease	Laryngeal cancer	[132]
miR-340	Decrease	Cervical cancer	[111]
miR-375	Decrease	Lung cancer (via JAK2/STAT1)	[133]
miR-383	Decrease	Cervical cancer	[111]
miR-424	Decrease	Ovarian cancer	[134]
miR-497-5p	Decrease	Renal cell carcinoma	[135]
miR-513	Decrease	Cholangiocytes	[96]
miR-519	Decrease	Pancreatic cancer	[136]
miR-570	Decrease	Gastric cancer	[137]
miR-873	Decrease	Breast cancer	[138]
miR-3609	Decrease	Breast cancer	[139]
miR-18a	Increase	Cervical cancer	[111]
miR-20	Increase	Colorectal cancer	[112]
miR-21	Increase	Colorectal cancer	[112]
miR-130b	Increase	Colorectal cancer	[112]
miR-135	Increase	Lung cancer	[113]
miR-3127-5p	Increase	Lung cancer (via STAT3)	[140]

MSI, microsatellite instability; CKS1B, CDC28 protein kinase regulatory subunit 1B; STAT3, Signal transducer and activator of transcription 3.

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
