# Peer review of "Spatial and Temporal Changes in PD-L1 Expression in Cancer: The Role of Genetic Drivers, Tumor Microenvironment and Resistance to Therapy"

_ijms, 2020, doi:10.3390/ijms21197139_

Round 1

Reviewer 1 Report

Ijms-918071

In this review the authors discuss the biological consequences of PD-L1 expression by various cell types within the tumor microenvironment, and the complex mechanisms that regulate PD-L1 expression at the genomic, transcriptomic and proteomic levels. Overall the manuscript is timely and interesting. The following comments needed to be addressed to improve the overall quality of the manuscript.

Specific comments:

  1. I suggest the authors to please take a look at the recent publication in Nature immunology by Shogo Kumagai et al about the significance of PD-1 expressing Tregs and add necessary discussion.
  2. Please add a section on how the PD-L1/PD-1 axis matters in context of innate and acquired resistance to immune checkpoint inhibitors.
  3. Please add a figure to showcase regulation of PD-L1

Regards,

Reviewer 2 Report

In this review, Elena Shklovskaya and Helen Rizos summarized the understanding of PD-L1 functions, focusing on the importance of PD-L1 expression on non-tumor cells vs. tumor cells, the regulation of PD-L1 expression and post-transcriptional modification, including most recent key studies.

The manuscript is clearly and well written. In general, I found this review interesting.

I would recommend only the following minor changes:

  • Page 2, line 59 : “Expression of PD-L1 in the tumor microenvironment can occur in the absence or presence of PD-L1 expression on tumor cells, and expression on both types can play a role in immune-mediated tumor control.“ – “in the tumor microenvironment” should be replaced by “on non-tumoral cells in the tumor microenvironment” as tumor cells are also part of the tumor microenvironment.
  • Page 2, line 69-71 : I would recommend deleting “In other animal studies, where direct interactions between T-cells and tumor cells were excluded, tumors could be eradicated efficiently by the host immune system that exclusively targeted the tumor microenvironment [17, 18]. “ as the link with previous text is confusing and the studies are not related to PD-L1.
  • Figure 1 legend: please add the type of cancer-related to this classification as this classification is not universal for all types of cancer.
  • Figure 3 legend : epithelial-to-mesenchymal transition (EMT) should be redefined as an abbreviation.

Page 7, line 288-289 : “ Importantly, tumor PD-L1 expression in EMT states is associated with poor responses to PD-1 blockade in multiple cancers” all references that are not directly supporting this statement should be deleted (in fact, some of those references are not correct).

  • Page 11, line 428-429: “Accordingly, pharmacological inhibition of these pathways decreased PD-L1 expression in the corresponding in vitro and in vivo models of cancer” - references that are not directly supporting this statement should be deleted. For instance, Tauriello et al demonstrated that the TGFβ inhibition is sufficient to confer susceptibility to anti-PD-1–PD-L1 checkpoint-based therapies but I am not sure that the PD-L1 expression was decreased after the TGFβ inhibition.
